# TOWARDS DYNAMIC EHR PHENOTYPING: A GENERATIVE CLUSTERING MODEL

## ABSTRACT

In healthcare, identifying clinical phenotypes—subgroups defined by specific clinical traits—is essential for optimizing patient care. The wealth of Electronic Health Record (EHR) information has fueled data-driven approaches to tackle this challenge. Unfortunately, the heterogeneity, multi-modality, and dynamic nature of EHR data pose significant hurdles. We propose DeepGC, a novel generative, clustering, outcome-sensitive end-to-end deep learning (DL) model for uncovering *dynamic phenotypes* within temporal EHR data. DeepGC leverages patient trajectories and outcomes to identify clinically meaningful phenotypes that evolve over time. Our generative model employs a dynamic sequential approach based on a Markovian Dirichlet distribution and Variational Auto-Encoders (VAEs), which is capable of providing insights into the evolution of patient phenotypes and health status. Preliminary evaluation indicates that DeepGC shows promise in identifying distinct and interpretable phenotypes, and outperforming existing benchmarks, particularly with regard to outcome sensitivity (3 % increase in F1). We also showcase the model's potential to yield valuable insights into the future evolution of patients' health status.

## 1 INTRODUCTION

In healthcare environments, *phenotyping*, or the precise identification of clinical phenotypes, subgroups of the general population characterized by distinct clinical traits, can be a valuable tool indispensable for optimizing patient care (Vogelmeier et al., 2018). A huge number of chronic conditions, for instance, Chronic Obstructive Pulmonary Disease (COPD), naturally present distinct phenotypes. It underlines the vital role of phenotype recognition in improving disease management through the refinement of treatment strategies guided by information in each phenotype (Adeloye et al., 2015). In practice, phenotypes encompass disparate clinical *aspects* collected over time, including observational measurements, treatment responses, and adverse event occurrences. Nevertheless, temporal Electronic Health Record (EHR) data, characterized by its heterogeneous nature, and exhibiting high dimensionality, multi-modality, non-stationarity, and data imbalance, etc. represent formidable hurdles not only for modeling but also for the analysis of clinical phenotypes (Conway et al., 2011).

Deep learning (DL) has emerged as a promising avenue for EHR data modeling (Rajkomar et al., 2018). However, current DL-based methods for phenotype identification primarily focus on observational data, often neglecting other crucial aspects such as patient outcomes, occurrence of adverse events, and clinical interventions. Existing frameworks also typically assume a phenotype is static over time, which is not necessarily aligned with the dynamic nature of patient health status in clinics. Finally, it is generally challenging to derive a meaningful clinical understanding of the derived phenotypes. We introduce a novel **Deep learning Generative outcome-based Clustering approach for dynamic phenotyping (DeepGC)** that aims to tackle the above limitations. Our method combines DL techniques with dynamic clustering to model patients' evolving physiological status over time. For a clinical application, DeepGC's generative approach enables the generation of unseen, future observation data, facilitating the exploration of future scenarios.

Our paper comprises the following sections: Section 2 reviews prior research in EHR time-series modeling, EHR phenotyping, clustering and generative models for multi-dimensional time-series data. In Section 3, we provide an in-depth exposition of DeepGC's generative and inference components,

as well as a derivation of the resulting optimization goal. Section 4 outlines the experimental setup, including a characterization of the datasets we considered as part of our experiments, a description of the evaluation pipeline and the obtained results. We discuss the results in Section 5. We conclude by highlighting the implications of our work and avenues for future research under Section 6.

## 2 BACKGROUND

In recent years, there has been a growing interest in utilizing data-driven approaches to phenotype medical cohorts through the analysis of multi-dimensional temporal EHR data (Davenport & Kalakota, 2019). Phenotyping involves the identification of patient subgroups with shared characteristics or disease trajectories, typically achieved through clustering techniques. This task, however, presents formidable challenges. Firstly, EHR data lacks labeled phenotypic information, making it necessary to rely on unsupervised learning methods for pattern discovery which are hard to validate and evaluate. Secondly, the multi-dimensional nature of EHR time-series data complicates the establishment of a suitable distance metric for assessing the similarity between sets of trajectories from distinct patients. Finally, clinical interpretation and validation of the resulting clusters and cluster assignments is extremely important, and it has been shown this is hard to address effectively within the EHR context (Xiao et al., 2018).

Standard clustering methods such as the K-Means algorithm (K-means, Lloyd (1982)), and Gaussian Mixture Models (GMM, Reynolds et al. (2009)) are not directly applicable to multidimensional time-series data, but simple variants have been introduced to this effect (Cuturi & Blondel, 2017). An extension to K-Means, Time-Series K-Means (Tavenard et al., 2020), was proposed based on applying the K-Means algorithm with a set of customized time-series distance metrics based on the idea of Dynamic Time Warping (Berndt & Clifford, 1994). DL methodologies, however, have largely led the performance landscape with respect to the modeling of multi-dimensional temporal EHR data and phenotyping. This is due to a variety of reasons but is largely driven by the lack of modeling capacity of simpler models to accurately identify relevant trends in the data (Shickel et al., 2018).

A key challenge in phenotype identification pertains to the incorporation of different *aspects* into the formation of the phenotypes. Different aspects need to be modeled very differently, and the resulting clusters are discordant (in fact, a good clustering for aspect A is typically not a good clustering for aspect B). Furthermore, some information might not be available until the end of a clinical admission (for example, the occurrence of adverse events), and so models must be capable of handling 'missing information' when deploying to non-training patient cohorts. To this end, the authors in Lee & Schaar (2020) propose AC-TPC to tackle phenotyping with both observation and occurrence of intra-admission events. T-Phenotype (Qin et al., 2023) is another alternative in a similar setting based on efficient representation learning in a frequency domain. Another approach, CAMELOT (Aguiar et al., 2022) was introduced to phenotype time-series data over observation and outcomes. However, the aforementioned methods display some limitations. Firstly, phenotype learning is applied to data within a fixed timeframe, which is not representative of the needs of a typical healthcare setting, where phenotypes, as representatives of underlying physiology, should be regularly updated and analyzed. Secondly, is it hard to characterize future physiology evolution based on the estimated phenotypes, again, due to the fixture of model analysis. Generative time-series methods, such as VRNN-GMM (Chung et al., 2015), employ techniques that can tackle both of these limitations due to their capability of generating observation data (including estimates of future observations). However, VRNN-GMM is limited due to its inability to identify clinical phenotypes that are outcome-sensitive. It also lacks temporal modeling of cluster assignments and the distribution of the mixture coefficients. Furthermore, it is not implemented in an end-to-end fashion, which is sub-optimal.

To that effect, we proposed a novel generative, clustering, outcome-sensitive model, DeepGC, to identify phenotypes in temporal clinical data. We also introduce a training algorithm to optimize a lower bound to the log-likelihood function. Our approach is capable of providing outcome-sensitive phenotypes in multiple aspects: (1) the dynamic nature of our model allows it to be applicable to realistic time-varying clinical scenarios; (2) the generation capabilities of DeepGC, together with the clustering nature, allow the extraction of clinically relevant knowledge and insight concerning individual patient's past and future health status.

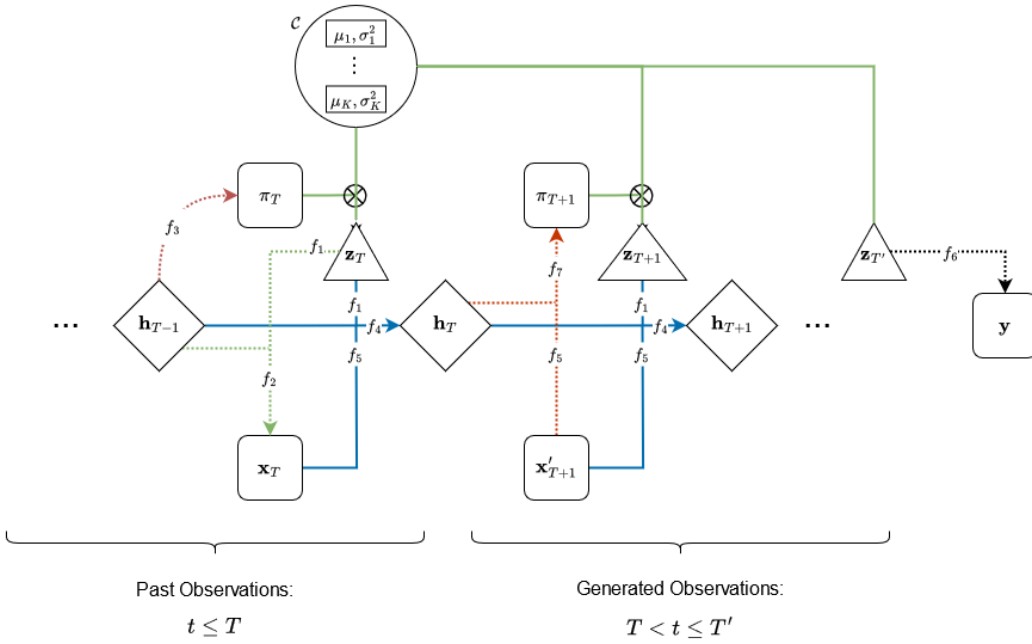

Figure 1: Sketch diagram for DeepGC given input data $\mathbf{x}_{\leq T}, \mathbf{y}$. At each time step, we model cluster assignment probability variables $\boldsymbol{\pi}_t$ via prior (red) and inference (orange) steps, observation data via the generation step (green), and we update the network's cell state (blue). Note that inference and generation steps occur at *every* time step - they are shown separately for ease of visualization. For $t > T$, generated data $\mathbf{x}'_t$ is used in lieu of $\mathbf{x}_t$ as we don't have access to 'future' data. Finally, the outcome is derived from the last estimated sample representation $\mathbf{z}_{T'}$ (black). Square boundaries denote probabilistic objects, while triangle boundaries encompass derived vectors. Finally, dashed lines represent probabilistic models.

## 3 OUR APPROACH

We propose a novel generative model called DeepGC for dynamically phenotyping patients via temporal EHR data. Our model characterizes patient phenotypes based on two distinct aspects: a) the trajectories of temporal observation sets, the *observation aspect*, and b) the empirical distribution of admission outcomes for patients in each phenotypic group, the *outcome aspect*. Note admission outcomes are obtained only at the end of a patient admission; further detail regarding the definition of the outcomes is present in Section 4.1. A sketch diagram of our model is shown in Figure 1.

DeepGC represents a joint probability model via a neural network based on a variational autoencoder (VAE) architecture coupled through time via a dynamic Markovian Dirichlet approach. At each time step, the latent space is modeled as a Gaussian Mixture Model (GMM) with a Dirichlet distribution prior, such that a patient representation is obtained as a weighted combination of cluster centroids (*i.e.* *phenotypic groups*). Across time, the Dirichlet prior is updated using a Recurrent Neural Network (RNN) under a Markovian assumption. Finally, DeepGC models the admission outcome with a VAE decoder framework applied to the last observable patient representation. Note that our model is end-to-end and optimization is done jointly across all components.

**Theoretical Formulation**   We consider a dataset $\mathcal{X} = \{\mathbf{x}_n\}_{n=1}^N$ of patient trajectories. We define: $N$ as the number of patients, $\mathbf{x} = \{\mathbf{x}_{n,t}\}_{t=1}^{T_n}$ as the set of observations for patient $n$, and $T_n$ the number of observations for patient $n$. Each $\mathbf{x}_{n,t} = [\mathbf{x}_{n,t}^1, ..., \mathbf{x}_{n,t}^f]$ denotes a set of measurements from $f$ clinical features taken at time $t$ and for patient $n$. A *trajectory* is the sequence of temporal observation sets collected with respect to the same feature. For simplicity, we denote $T := \max_n \{T_n\}$ to be the maximum number of time steps.

Separately, we have a dataset of patient outcomes $\mathcal{Y} = \{\mathbf{y}_n\}_{n=1}^N$ for each patient. We assume outcomes are one-hot encoded for a total of $C > 1$ possibilities. In general, due to our global application objective, outcomes represent the occurrence (or lack thereof) of adverse events that occur no earlier than a fixed window of time after the last observation. This is to avoid modeling events taking place concurrently as this is not useful information for medical correction or intervention.

## 3.1 MODEL ARCHITECTURE

Given input data $\mathbf{X}$, DeepGC iteratively computes a temporal sequence of (a) *cluster probability assignment vectors*, or simply, *assignment probabilities*, $\boldsymbol{\pi}_t \in \Delta_{K-1}$, and (b) generated observation samples, $\hat{\mathbf{x}}_t$, using a Dirichlet Markovian approach (detailed below). Here, we use the notation $\Delta_{K-1} := \{\mathbf{v} \in \mathbb{R}^K : \|\mathbf{v}\|_1 = 1, \mathbf{v} \geq \mathbf{0}\}$, and the hyperparameter $K$ denotes the number of phenotypes/clusters. Assignment probabilities are used to estimate patient representations, $\mathbf{z}_t$, based on a weighted average with the set of phenotype centroids, $\mathcal{C} = \{\mathbf{c}_1, ..., \mathbf{c}_K\}$. The cluster centroids are randomly initialized and then optimized by gradient descent, as per the other model parameters. Importantly, and distinct from the existing literature, DeepGC 'looks into the future' via forward generation of unseen observations. This is achieved by continuing to generate observations after time step $T$ and iteratively leveraging the generated data to compute assignment probabilities and vice-versa. The number of such extra time steps generated is also a pre-defined hyperparameter. Finally, the patient representation obtained at the last time step in the model passes through a set of neural network layers to compute the outcome of the patient admission. DeepGC models the joint probability:

$$p(\boldsymbol{\pi}_{\leq T}, \mathbf{x}_{\leq t}, \mathbf{y}) = p(\mathbf{y} \mid \mathbf{x}_{\leq T}, \boldsymbol{\pi}_{\leq T}) \prod_{t=1}^T p(\mathbf{x}_t \mid \boldsymbol{\pi}_{\leq t}, \mathbf{x}_{<t}) p(\boldsymbol{\pi}_t \mid \boldsymbol{\pi}_{<t}, \mathbf{x}_{<t}) \tag{1}$$

by dynamically modeling the two inner probability terms, denoted as **Generating Step** and **Prior Step**, respectively, akin to an RNN approach. The outer term is referred to as the **Outcome Step**. Furthermore, we also implement an **Inference Step** which computes the assignment probabilities, and it approximates the intractable posterior $p(\boldsymbol{\pi}_t \mid \boldsymbol{\pi}_{<t}, \mathbf{x}_{\leq t})$. We make two added assumptions: (a) at each time step $t$, the previous cell state, $\mathbf{h}_{t-1}$ in an RNN is representative of $(\boldsymbol{\pi}_{<t}, \mathbf{x}_{<t})$, (b) the outcome conditional probability is solely dependent on the last estimated representation $\mathbf{z}_{T'}$ (recall $T' > T$ as DeepGC generates future unseen data). A pseudo-algorithmic description of our model is shown in Algorithm 1. DeepGC includes a set of feed-forward neural networks, which are leveraged to model the generating, prior, state update, outcome, and inference components. For ease of notation, we will use $f_i(\mathbf{x})$ to denote the function encoded by the $i-$th corresponding feed-forward component (in no particular order).

**Generating Step** Observation data are modeled as a normal distribution with mean and variance obtained as the outputs of a feedforward network (akin to a VAE decoder). Given $\mathbf{h}_{t-1}, \boldsymbol{\pi}_t, \mathcal{C}$, the phenotype-derived patient representation is computed, $\mathbf{z}_t = \sum_{c_k \in \mathcal{C}} \pi_t^k c_k$, then the relevant features are extracted, $f_1(\mathbf{z}_t)$, and we concatenate this embedding with the cell-state to estimate the distribution parameters, $[\boldsymbol{\mu}, \boldsymbol{\sigma}^2] = f_2(\mathbf{h}_{t-1}, f_1(\mathbf{z}_t))$. Input data are generated according to $\mathbf{x}_t^{est} \sim \mathcal{N}(\boldsymbol{\mu}, \boldsymbol{\sigma}^2 I_f)$.

**Prior Step** We employ the Dirichlet Markovian approach to model the prior. Given $(\boldsymbol{\pi}_{<t}, \mathbf{x}_{<t})$, $\boldsymbol{\pi}_t$ is distributed according to a Dirichlet distribution with parameter $\boldsymbol{\alpha}_t^p$. Together with the assumption (a) described above, we model $\boldsymbol{\pi}_t \mid \mathbf{h}_{t-1} \sim \mathcal{D}(\boldsymbol{\alpha}_t^p)$, and $\boldsymbol{\alpha}_t^p = f_3(\mathbf{h}_{t-1})$.

**State Update Step** We model $\mathbf{h}_t \mid (\boldsymbol{\pi}_{<t}, \mathbf{x}_{<t}) \sim \mathbf{h}_t \mid (\boldsymbol{\pi}_t, \mathbf{x}_t, \mathbf{h}_{t-1}) = f_4(\mathbf{h}_{t-1}, f_1(\mathbf{z}_t), f_5(\mathbf{x}_t))$, where $f_5$ is a feature extractor for the input space, and $\mathbf{z}_t$ is derived from the Generating step.

**Outcome Step** We use a Categorical distribution whose parameter is the output of a feed-forward network, as well as the assumption (b) to model the outcome, so that $\mathbf{y} \mid \boldsymbol{\pi}_{<T}, \mathbf{x}_{<T} \sim \hat{\mathbf{y}} \mid \boldsymbol{\pi}_{T'} \sim \text{Cat}(f_6(\mathbf{z}_{T'}))$. We reinforce the idea that we use the latest estimated representation $\mathbf{z}_{T'}$, which occurs *after* the last observed time step via forward data generation.

**Inference Step** The posterior $p(\boldsymbol{\pi}_t \mid \boldsymbol{\pi}_{<t}, \mathbf{x}_{\leq t})$ is approximated by DeepGC according to a Dirichlet distribution $\boldsymbol{\pi}_t \mid \boldsymbol{\pi}_{<t}, \mathbf{x}_{\leq t} \sim \boldsymbol{\pi}_t \mid \mathbf{h}_{t-1}, \mathbf{x}_t \sim \mathcal{D}(\boldsymbol{\alpha}_t^i)$, with $\boldsymbol{\alpha}_t^i = f_7(\mathbf{h}_{t-1}, f_5(\mathbf{x}_t))$. We use $q$ to represent the probability distribution function encoded by the inference step of the model.

---

**Algorithm 1** Pseudo-algorithmic description of DeepGC

---

**Require:** Temporal data $X = (\mathbf{x}_1, ..., \mathbf{x}_T)$, outcome $\mathbf{y}$, centroids $\mathcal{C}$ hyper-parameters $T', K$.
**Ensure:** Latent variables $\boldsymbol{\pi}_t$, generated observations $\mathbf{x}_t^{est}$, for $t = 1, ..., T'$, and prediction $\hat{\mathbf{y}}$.
 1: Initialize cell state $h_0$
 2: **for** $t = 1$ to $T$ **do**
 3:     sample $\boldsymbol{\pi}_t \sim q(\boldsymbol{\pi}_t \mid \mathbf{x}_t, \mathbf{h}_{t-1})$
 4:     $x_{\text{est}}(t) \sim p(x_{\text{est}}(t) | \boldsymbol{\pi}_t, \mathbf{h}_{t-1})$
 5:     Update cell state: $\mathbf{h}_t \leftarrow f_4(\mathbf{h}_{t-1}, f_1(\mathbf{z}_t), f_5(\mathbf{x}_t))$
 6: **end for**
 7: **Forward variable generation:**
 8: **for** $t = T + 1$ to $T'$ **do**
 9:     sample $\boldsymbol{\pi}_t \sim q(\boldsymbol{\pi}_t \mid x_{\text{est}}(t), \mathbf{h}_{t-1})$
10:     $x_{\text{est}}(t) \sim p(x_{\text{est}}(t) | \boldsymbol{\pi}_t, \mathbf{h}_{t-1})$
11:     Update cell state: $\mathbf{h}_t \leftarrow f_4(\mathbf{h}_{t-1}, f_1(\mathbf{z}_t), f_5(x_{\text{est}}(t)))$
12: **end for**
13: **Outcome Prediction:**
14: sample $\hat{\mathbf{y}} \leftarrow \hat{\mathbf{y}} \mid \boldsymbol{\pi}_{T'}$
15: **return** $\boldsymbol{\pi}_t, \mathbf{x}_t^{est}$, for $t = 1, ..., T'$, and $\hat{\mathbf{y}}$

---

## 3.2 MODEL OPTIMIZATION

Ideally, we would train our model to optimize the log-likelihood:

$$p(\mathbf{x}_{\leq T}, \mathbf{y}) = \int p(\mathbf{x}_{\leq T}, \mathbf{y}, \boldsymbol{\pi}_{\leq T}) d\boldsymbol{\pi}_{\leq T} = \int p(\mathbf{y} \mid \mathbf{h}_T) \prod_{t=1}^{T} p(\mathbf{x}_t \mid \boldsymbol{\pi}_t, \mathbf{h}_{t-1}) p(\boldsymbol{\pi}_t \mid \mathbf{h}_{t-1}) d\boldsymbol{\pi}_{\leq T}$$

(2)

where we use the decomposition in Equation 1, as well as our modeling assumptions. This integral is generally intractable, and so cannot easily be evaluated directly. Several methods have been proposed to approximate the likelihood directly based on sampling such as Monte Carlo (Luengo et al. (2020)), or Markov Chain Monte Carlo (Chib, 2001). However, such methods are computationally expensive. Therefore, we propose a Variational Inference (VI) approach using the Evidence Lower BOund (ELBO) method (Blei et al., 2017).

The ELBO method makes use of a family of amortized distributions (distributions that approximate the posterior) to derive a lower bound to the log-likelihood, $\mathcal{L}$, which is maximized. Under certain conditions, it can be shown that $\mathcal{L}$ maximization is equivalent to maximizing the similarity to the posterior and approximating its distribution (Rezende et al., 2014). We define our family to be those represented by our inference step model - note that $q(\boldsymbol{\pi}_{\leq T} \mid \mathbf{x}_{\leq T}, \mathbf{y}) = \prod_t q(\boldsymbol{\pi}_{\leq t} \mid \mathbf{x}_t, \mathbf{h}_{t-1})$, so we can extend our inference step to a probability distribution over all latent variables $\boldsymbol{\pi}$. The ELBO for our model is then given by:

**Lemma 1.** *The ELBO lower bound for the generative model represented by DeepGC is given by:*

$$\mathcal{L} = \mathbb{E}_{\boldsymbol{\pi}_{\leq T} \sim q(\boldsymbol{\pi}_{\leq T} | \mathbf{x}_{\leq T})} \Big[ \log p(\mathbf{y} \mid \boldsymbol{\pi}_{\leq T}) + \sum_{t=1}^{T} \big( \log p(\mathbf{x}_t \mid \boldsymbol{\pi}_t, \mathbf{h}_{t-1}) - $$
$$D_{\text{KL}}(q(\boldsymbol{\pi}_t \mid \mathbf{x}_t, \mathbf{h}_{t-1}) \| p(\boldsymbol{\pi}_t \mid \mathbf{h}_{t-1}))) \Big]$$

(3)

*Proof.* The full derivation is described in the Appendix Section A.1.                    □

We train our model to maximize the log-likelihood $\mathcal{L}$. The intuitive contribution of each term in the ELBO expression can be explained as follows: firstly, given uncertainty about the latent variables $\boldsymbol{\pi}_{<T}$, we average over the inference distribution; $\log p(\mathbf{y} \mid \boldsymbol{\pi}_{\leq T})$ encourages the modeling of the outcome aspect of the resulting phenotypes; $\log p(\mathbf{x}_t \mid \boldsymbol{\pi}_t, \mathbf{h}_{t-1})$ quantifies the quality of data generation; finally, the Kullback-Leibler (KL) divergence, denoted as $D_{\text{KL}}$, is a measure of the quality

of our posterior approximation. However, the optimization of our neural network concerning Equation 3 presents notable challenges. These challenges revolve around two core issues. Firstly, the intricate computation of $D_{KL}$ between the inference and prior distributions stemming from their inherent complexity. We show we can simplify $D_{KL}$ using properties of the Dirichlet distribution, reducing its dependence solely on the distribution parameters in a differentiable fashion (full derivation in the Appendix Section A.2). Secondly, the computation of the expectation in a manner that enables effective backpropagation of gradients through the network parameters. It has been shown that sampling through the $q$ distribution is a strong estimator with low variance for estimating the ELBO Rezende et al. (2014). However, it is challenging to sample from a Dirichlet distribution while allowing backpropagation of gradients through the distribution parameter. We use a Dirichlet sampling method based on (a) a Dirichlet reparameterization into independent Gamma-distributed variables with shape $\pi_i$ and rate 1, and (b) an approximate parameter-differentiable gamma sampling mechanism (Joo et al., 2020) based on the inverse transform sampling theorem (Devroye, 1986). A description of our algorithm is shown in Algorithm 2 and a full derivation of its correctness is present in the Appendix Section A.2

---

**Algorithm 2** Proposed Dirichlet Sampling Algorithm

---

**Require:** Number of components $K$, parameter $\boldsymbol{\alpha}$
**Ensure:** Dirichlet sample $\mathbf{v} \sim \mathcal{D}(\boldsymbol{\alpha})$
 1: Initialize output of size $K$: $\mathbf{v} \leftarrow [0, 0, \ldots, 0]$
 2: **for** $k = 1$ to $K$ **do**
 3:    Sample random variable $u_k \sim \text{Unif}(0, 1)$
 4:    $g_k \leftarrow (z u_k \Gamma(u_k))^{1/u_k}$
 5: **end for**
 6: Set $\mathbf{v}_k \leftarrow \frac{\mathbf{g}_k}{\sum_{k=1}^{K} \mathbf{g}_k}$
 7: **return** $\mathbf{v}$

---

### 3.3 MODEL TRAINING

For the purposes of our experiments, $T'$ was picked as $2 * T$, i.e., we generate a new period of future data that is used to estimate the outcome. All feed-forward neural network components of our model were implemented with 2 hidden layers of 30 nodes each, and a hyperbolic tangent activation function. For computing parameters of Dirichlet distributions, a Rectified Linear Unit activation was further applied to ensure all vector components were non-negative. Glorot initialization (Glorot & Bengio, 2010) was used for all neural network weights, with the exception of bias weights, which are initialized according to a standard normal distribution. Our model was implemented in Python using the Pytorch (Paszke et al., 2017), Scikit-learn (Fabian et al., 2011) and NumPy (Harris et al., 2020) packages, with environment specifications indicated in our online repository, and all experiments were run with 1 NVIDIA Tesla V100 GPU, and 8 CPUs Intel(R) Xeon(R) Gold 6246 @3.30 GHz.

## 4 EVALUATION AND RESULTS

### 4.1 DATA AND PRE-PROCESSING

For our experiments, we utilized two distinct medical datasets, each offering unique insights into the healthcare domain. **HAVEN** (HAVEN project, REC reference: 16/SC/0264 and Confidential Advisory Group reference 08/02/1394) comprises proprietary secondary care records from a United Kingdom hospital, while **MIMIC-IV-ED** (named MIMIC hereafter, Johnson et al. (2023)) is a publicly available dataset from a United States Emergency Department (ED). HAVEN primarily informed model development and clinical applications, while MIMIC provided evidence of model generalizability and reproducibility. Both datasets share similar characteristics, including EHR heterogeneity, data multimodality, and significant class imbalance among multiple outcome categories.

**HAVEN** The HAVEN Database consists of clinical observations, such as vital signs and laboratory measurements, collected from March 2014 to March 2018 at Oxford University Hospitals NHS Foundation Trust. The cohort focused on patients at risk of developing Type-II Respiratory Failure

(T2RF) during hospitalization, following the protocol in Pimentel et al. (2019). After filtering for observation availability, we averaged data over 4-hour blocks. These values were determined based on clinical insight concerning observation recording and clinical staff visit frequency. Outcomes include death ('Death'), cardiac arrest ('Cardiac'), unplanned ICU admission ('ICU'), based on the first occurrence of an adverse event, and healthy discharge from hospital ('Healthy'), otherwise. Post-processing yielded over 100,000 patient trajectories across 4,266 unique hospital admissions, with a severe class imbalance: 86.8% (Healthy), 10.3% (Death), 1.8% (ICU) and 1.1% (Cardiac). Models were trained using observations between 48 and 72 hours prior to the onset of the event time.

**MIMIC-IV-ED**  MIMIC-IV-ED documents ED admissions at the Beth Israel Deaconess Medical Center, with records of vital signs, triage data, medications, and patient hospital journeys. Pre-processing paralleled HAVEN, selecting patients by observation availability, and averaging data over 1-hour blocks. We removed patients with an Emergency Severity Index (ESI, Gilboy et al. (2011)) score of 1 or 5, as these potentially represented extreme cases (too ill or not ill enough). Outcomes within 12 hours of admission include patient demise ('Death'), ICU admission ('ICU'), discharge ('Discharge'), and presence in a non-intensive ward ('Ward'). Like HAVEN, MIMIC exhibits class imbalance: 81.06% (Ward), 16.53% (ICU), 2.11% (Discharge), 0.30% (Death). Models were trained on observations corresponding to the last 6 hours of the ED admission.

Both datasets underwent preprocessing based on clinical knowledge and empirical validation as in Aguiar et al. (2022). Our analysis assessed cluster phenotypes, considering dissimilarities in trajectory data ('observation aspect') and outcome distributions ('outcome aspect'). The post-processed cohort was split into training (60%) and testing (40%) data based on patient admissions. Deep learning models further divided the training data into training (60%) and validation (40%) subsets. Features were min-max normalized, and missing values were imputed based on observed values, with static features introduced as constant time-series. Normalization and imputation parameters derived from training data were applied to test data (and validation data for deep learning models).

## 4.2 Performance Evaluation

In the realm of temporal EHR data, DeepGC is tailored to the critical task of identifying patient phenotypes. To comprehensively assess its performance, we conducted benchmark comparisons with classical clustering models like Time-Series K-Means (TSKM) and GMM, advanced phenotyping methods such as CAMELOT, and AC-TPC, as well as VRNN-GMM, a deep learning model well-regarded for clustering multi-dimensional structured time-series data. Our evaluation focused on the models' performance using the final available time step of the input data.

We examined the learned clusters from two pivotal perspectives: observation and outcome. For the observation aspect, we evaluated how effectively the models clustered patient trajectories, employing well-established metrics such as Silhouette score (SIL, Rousseeuw (1987)), Davies-Bouldin Index (DBI, Davies & Bouldin (1979)), and Variance Ratio Criterion (VRI, Caliñski & Harabasz (1974)). Assessing the outcome aspect presented a unique challenge, given that each cluster corresponded to a distinct outcome distribution. We are interested in identifying different outcome distributions over each cluster, not necessarily identifying clusters associated with a single outcome, as the latter show very poor separation at the observation aspect. We used a *separate task as a proxy to evaluate this aspect* - namely the more straightforward outcome prediction task. Purely unsupervised models without an outcome output were extended to classification methods by assigning each patient to the training outcome distribution of the cluster they belong to. Higher scores signified that the clusters effectively captured the relevant outcomes and exhibited better separation. We evaluated prediction performance using multi-class Macro-average AUROC, F1-score (F1), Recall, and Normalized Mutual Information (NMI).

Furthermore, to underscore the practical relevance of our model, we compared its performance with standard classifiers like Support Vector Machines (SVM), XGBoost (XGB), and NEWS2 and ESI, clinical benchmarks for HAVEN and MIMIC, respectively. To handle time-series data, we considered 2 natural extensions for both SVM and XGB. Firstly, we considered each (clinical feature, time step) pair as a separate feature passed to the classifier models. Alternatively, we built an ensemble classifier, building individual classifiers for each particular feature's trajectory data. The best result is shown in Table 1. With NEWS2, we built a classifier based on the NEWS2 score of the last training observation set. The comparison with standard classifiers demonstrated our model's ability to identify

| Dataset | Model | AUROC | F1 | Recall | Precision | SIL | VRI | DBI |
|---------|-------|-------|-----|--------|-----------|-----|-----|-----|
| **HAVEN** | TSKM | 0.53 (.03) | 0.22 (.02) | 0.25 (.02) | 0.25 (.01) | **0.48**(.10) | **417**(8.20) | **1.11**(.20) |
| | GMM | 0.51 (.01) | 0.27 (.01) | 0.26 (.01) | 0.27 (.03) | 0.00 (.00) | 98.7 (11.32) | 3.56 (.28) |
| | CAMELOT | **0.65**(.02) | 0.30 (.03) | 0.32 (.01) | 0.29 (.02) | 0.14 (.05) | 157.2 (27.0) | 2.37 (.36) |
| | VRNN-GMM | 0.55 (.03) | 0.26 (.01) | 0.27 (.00) | 0.28 (.01) | 0.11 (.03) | 103.2 (4.56) | 3.13 (.87) |
| | **DeepGC** | **0.65**(.03) | **0.32**(.02) | **0.33**(.01) | 0.29(.03) | 0.22 (.04) | 189.9 (29.2) | 1.93 (.15) |
| | SVM | 0.56 (.01) | 0.25 (.00) | 0.26 (.01) | 0.26 (.00) | - | - | - |
| | XGB | 0.61 (.03) | 0.29 (.01) | 0.29 (.01) | **0.30**(.01) | - | - | - |
| | NEWS2 | 0.53 | 0.26 | 0.27 | 0.25 | - | - | - |
| **MIMIC** | TSKM | 0.63 (.01) | 0.27 (.01) | 0.30 (.01) | 0.26 (.01) | **0.16**(.03) | **471.9**(80.7) | **1.59**(.10) |
| | GMM | 0.61 (.05) | 0.25 (.01) | 0.28 (.03) | 0.36 (.01) | 0.03 (.06) | 143.4 (138.2) | 4.41 (1.26) |
| | CAMELOT | 0.63 (.03) | 0.29 (.02) | 0.32 (.03) | 0.30 (.02) | 0.11 (.05) | 302.5 (25.1) | 2.77 (.20) |
| | VRNN-GMM | 0.55 (.01) | 0.18 (.01) | 0.27 (.00) | 0.26 (.01) | 0.02 (.03) | 41.6 (6.5) | 3.8 (.62) |
| | **DeepGC** | **0.65**(.02) | **0.35**(.01) | **0.34**(.01) | 0.36 (.01) | 0.10 (.01) | 316.7 (15.1) | 2.65 (.24) |
| | SVM | 0.61 (.06) | 0.32 (.00) | 0.32 (.00) | **0.41**(.00) | - | - | - |
| | XGB | 0.56 (.04) | 0.32 (.01) | **0.34**(.03) | 0.40 (.01) | - | - | - |
| | ESI | 0.46 (.00) | 0.22 (.00) | 1.00 (.00) [*] | 0.25 (.00) | - | - | - |

Table 1: Performance scores of DeepGC and benchmarks on HAVEN and MIMIC datasets across observation and outcome aspects. We show average metric performance and standard deviation (in parenthesis) over the same set of 10 seeds. Dashed lines are used whenever the metric is not applicable to the corresponding model. Note ESI is unable to identify all classes, hence why it results in perfect Recall.

clinically meaningful patient phenotypes based on outcomes. Due to the imbalanced nature of our data, we consider only those experiments where all classes are predicted **at least once**. Table 1 shows the scores obtained by our model and benchmarks across both datasets.

We used a grid search (Bergstra & Bengio, 2012) approach to hyperparameter tuning, and we used Macro-average F1 as the main goal metric due to class imbalance. We used 5-fold cross-validation to tune hyper-parameters, and then evaluated performance on the hold-out test set. Results were averaged over 10 fixed seeds. Throughout the process, we adhered to the Occam's Razor (Hamilton, 1861) approach for hyperparameter optimization, selecting integer hyperparameter values that did not yield a statistically significant improvement in performance when further increased. We also ensured that the models selected could correctly identify at least one patient from each outcome class, effectively addressing the challenge posed by class imbalance in the data.

## 5 Discussion

Compared with all other benchmarks, our model shows an increase performance in phenotyping clusters (Table 1), particularly improving performance related to the outcome aspect (at least 2 % increase in average F1 score over both datasets, and improvements over all benchmarks in average F1 score and average Recall score, with the exception of XGBoost on the MIMIC Dataset). With the exception of TSKM, our model also outperforms all benchmarks with respect to the observation aspect, as can be seen by second best performance in clustering metrics SIL, DBI and VRI in both datasets. TSKM shows better clustering performance, but this is expected for three reasons. Firstly, TSKM learns cluster decision boundaries. Secondly, clustering metrics such as SIL, DBI and VRI have a natural bias towards convexity in cluster formation due to the distance functions used in their formulation. Finally, TSKM clusters directly on the input space, while DL models cluster on a latent space. Ideally, we would be able to compare cluster formation in latent space with the TSKM cluster formation, but this is not a trivial task due to the different space designs. We stress that DeepGC outperforms all other benchmarks across all metrics and datasets, however, except SIL performance on the MIMIC dataset, where CAMELOT displays a similar performance to DeepGC.

Our proposed model demonstrates superior capabilities in extracting intricate patterns from complex and heterogeneous electronic health record (EHR) data compared to prior models. Notably, the model excels in predictive tasks, even in the presence of a clustering bottleneck, where patient outcomes are determined based on assigned clusters rather than tailored to individual input data. While alternative models may potentially outperform ours in direct outcome prediction tasks, they could face challenges related to robustness, input sensitivity, and the identification of clinically relevant trends. Consequently, these alternatives may provide outcome predictions for new admissions but

may lack a comprehensive understanding of the underlying processes, preventive strategies, and data aggregation capabilities that our model offers.

## 6 CONCLUSION

In this paper, we introduce a novel generative deep learning model tailored for the identification of distinctive clusters within temporal EHR data, with a focus on phenotypic characteristics. Our proposed model leverages a sequential variational approach that serves several crucial purposes: a) modeling the probabilistic assignment of clusters at each time step, b) generating observational data over time, and c) forecasting patient outcomes. Through extensive experimentation on two independent datasets, including a publicly available dataset, we have observed promising outcomes regarding the distinguishability of clusters and the accuracy of outcome predictions. The integration of a deep learning-based probabilistic model significantly enhances the capacity to capture relevant representations and form meaningful clusters. Furthermore, our approach provides valuable insights into the dynamic evolution of cluster probability assignments during a patient's health journey.

It is worth emphasizing that further scrutiny is essential to assess the effectiveness of our generative approach. Data generation holds increasing importance in the healthcare domain, and we plan to explore the pertinence of our generative framework and delve into the generated observations, which hold particular significance due to data access constraints in healthcare (while generated data remains accessible). Preliminary qualitative analysis showcases our model is capable of generating relevant EHR data. Our upcoming research will focus on evaluating the robustness and sensitivity of the generated data to noise. Additionally, we intend to conduct an in-depth analysis of temporal cluster assignments, investigate the driving factors behind cluster transitions over time, and devise accurate evaluation methodologies for learned cluster assignments.

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

## A MATHEMATICAL DERIVATIONS

In this section, we provide more detail into some of the formulae obtained within our main text.

### A.1 ELBO

**Lemma 2.** *The ELBO lower bound for the generative model represented by DeepGC is given by:*

$$\mathcal{L} = \mathbb{E}_{\boldsymbol{\pi}_{\leq T} \sim q(\boldsymbol{\pi}_{\leq T} | \mathbf{x}_{\leq T})} \big[ \log p(\mathbf{y} \mid \boldsymbol{\pi}_{\leq T}) + \sum_{t=1}^{T} \big( \log p(\mathbf{x}_t \mid \boldsymbol{\pi}_t, \mathbf{h}_{t-1}) - D_{\mathrm{KL}}(q(\boldsymbol{\pi}_t \mid \mathbf{x}_t, \mathbf{h}_{t-1}) \| p(\boldsymbol{\pi}_t \mid \mathbf{h}_{t-1}))) \big]$$

*Proof.* To derive the ELBO, we start with Equation 1:

$$p(\mathbf{x}_{\leq T}, \mathbf{y}) = \int p(\mathbf{x}_{\leq T}, \mathbf{y}, \boldsymbol{\pi}_{\leq T}) d\boldsymbol{\pi}_{\leq T}$$

Define a new set of variables $\mathbf{x}'$ via $\mathbf{x}'_t = \mathbf{x}_t$ for $t < T$, and $\mathbf{x}'_T = [\mathbf{x}_T, \mathbf{y}]$, so our log-likelihood can be written as $\int p(\mathbf{x}'_{\leq T} \mid \boldsymbol{\pi}_{\leq T}) d\boldsymbol{\pi}_{\leq T}$. We can naturally extend the probability distribution given by $q$ to variables $\mathbf{x}'_T$ by using DeepGC's assumption (a) and (b). Denote the probability density function of the extension with $q'$. In this new setting we can use the work of Chung et al. (2015) to obtain the ELBO equation:

$$\mathcal{L} = \mathbb{E}_{\boldsymbol{\pi}_{\leq T} \sim q(\boldsymbol{\pi}_{\leq T} | \mathbf{x}'_{\leq T})} \Big[ \sum_{t=1}^{T} \big( \log p(\mathbf{x}'_t \mid \boldsymbol{\pi}_t, \mathbf{h}_{t-1}) - D_{\mathrm{KL}}(q(\boldsymbol{\pi}_t \mid \mathbf{x}'_t, \mathbf{h}_{t-1}) \| p(\boldsymbol{\pi}_t \mid \mathbf{h}_{t-1}))) \Big]$$

For $t < T$, all expressions are unaltered when replacing $\mathbf{x}$ and $\mathbf{x}'$ since $\mathbf{x}_t = \mathbf{x}'_t$. For $t = T$, we look at each individual term. First, the prior step remains altered as it does not depend on any input observation. Similarly, we may re-arrange the terms in the $q(\boldsymbol{\pi}_t \mid \mathbf{x}'_t, \mathbf{h}_{t-1})$ term of the KL divergence, to conclude that the KL divergence expression is the same for all time steps. On the other hand:

$$\log p(\mathbf{x}'_T \mid \boldsymbol{\pi}_T, \mathbf{h}_{T-1}) = \log p(\mathbf{x}_T, \mathbf{y} \mid \boldsymbol{\pi}_T, \mathbf{h}_{T-1}) = \log p(\mathbf{x}_T \mid \boldsymbol{\pi}_T, \mathbf{h}_{T-1}) + \log p(\mathbf{y} \mid \boldsymbol{\pi}_T, \mathbf{h}_{T-1})$$

where we used conditional independence given $\boldsymbol{\pi}_T$. Separating the second term outside the summation, and re-arranging, we obtain:

$$\mathcal{L} = \mathbb{E}_{\boldsymbol{\pi}_{\leq T} \sim q(\boldsymbol{\pi}_{\leq T} | \mathbf{x}_{\leq T})} \Big[ \log p(\mathbf{y} \mid \boldsymbol{\pi}_{\leq T}) + \sum_{t=1}^{T} \big( \log p(\mathbf{x}_t \mid \boldsymbol{\pi}_t, \mathbf{h}_{t-1}) -$$
$$D_{\mathrm{KL}}(q(\boldsymbol{\pi}_t \mid \mathbf{x}_t, \mathbf{h}_{t-1}) \| p(\boldsymbol{\pi}_t \mid \mathbf{h}_{t-1}))) \Big]$$

as required. □

## A.2 Computing the ELBO

It is not trivial to compute the ELBO (Equation 3) in a differentiable fashion. First, we show how to simplify the KL divergence term as a function of the distribution parameters.

**Lemma 3.** *The KL divergence between two Dirichlet distributions $Z_1$ and $Z_2$ with probability density functions $p_1(\mathbf{v}|\boldsymbol{\alpha})$ and $p_2(\mathbf{v}|\boldsymbol{\beta})$, respectively, is given by:*

$$D_{\mathrm{KL}}(Z_1 \| Z_2) = \log \left( \frac{\Gamma(\alpha_0)}{\Gamma(\beta_0)} \right) + \sum_i \log \left( \frac{\Gamma(\alpha_i)}{\Gamma(\beta_i)} \right) + \sum_i (\alpha_i - \beta_i) \left( \psi(\alpha_i) - \psi(\alpha_0) \right)$$

*Where:*
- $\boldsymbol{\alpha} = (\alpha_1, \alpha_2, \ldots)$, *and* $\boldsymbol{\beta} = (\beta_1, \beta_2, \ldots)$.
- $\alpha_0 = \sum_i \alpha_i$ *and* $\beta_0 = \sum_i \beta_i$.
- $\Gamma(\cdot)$ *is the gamma function.*
- $\psi(\cdot)$ *is the digamma function.*

*Proof.* The proof uses general properties of the Dirichlet distribution (Ng et al., 2011), specifically:

- The marginal distributions of $Z_1$ are given by Beta distributions: $Z_1^j \sim \mathrm{Beta}(\alpha_j - \alpha_0 - \alpha_j)$;

- The negative entropy of the marginals can be written in terms of the digamma function: $\mathbb{E}[-\log Z_i^j] = \psi(\alpha_0) - \psi(\alpha_j)$.

Then, we have:

$$D_{\mathrm{KL}}(Z_1 \| Z_2) = -\int_{\Delta_{d-1}} \log p_{Z_1}(z) \frac{p_{Z_2}(z)}{p_{Z_1}(z)} dz = \mathbb{E}_{z \sim p_{Z_1}} \left[ \log \frac{p_{Z_1}}{p_{Z_2}} \right]$$
$$= \mathbb{E}_{z \sim p_{Z_1}} \left[ \log p_{Z_1} - \log p_{Z_2} \right]$$
$$= \mathbb{E}_{z \sim p_{Z_1}} \left[ \log \Gamma(\alpha_0) - \sum_i \log \Gamma(\alpha_i) + \sum_i (\alpha_i - 1) \log(Z_1^i) - \right.$$
$$\left. \log \Gamma(\beta_0) + \sum_i \log \Gamma(\beta_i) - \sum_i (\beta_i - 1) \log(Z_1^i) \right]$$
$$= \log \Gamma(\alpha_0) - \log \Gamma(\beta_0) - \sum_i \left[ \log \Gamma(\alpha_i) - \log \Gamma(\beta_i) \right] +$$
$$\sum_i (\alpha_i - \beta_i) \mathbb{E}_{z \sim Z_1^i} \left[ \log Z_1^i \right]$$

and the result follows from the Dirichlet marginal entropy formula.

□

Lemma 3 allows simplification of the $D_{\mathrm{KL}}$ with respect to the distribution parameters, which are network outputs. We now show how we can sample through a Dirichlet random variable to allow backpropagation of gradients. We start with the following lemma:

**Lemma 4.** *A Dirichlet-distributed random vector* $\mathbf{v}$ *with dimensionality* $K$ *and concentration parameter* $\boldsymbol{\alpha} = (\alpha_1, \alpha_2, \ldots, \alpha_K)$ *can be expressed as a normalized vector of independent gamma-distributed random variables:*

$$\mathbf{v} = \left( \frac{v_1}{\sum_{i=1}^{K} v_i}, \frac{v_2}{\sum_{i=1}^{K} v_i}, \ldots, \frac{v_K}{\sum_{i=1}^{K} v_i} \right) \sim \mathcal{D}(\boldsymbol{\alpha}) \tag{4}$$

*Where* $v_i$ *is Gamma*$(\alpha_i, 1)$ *distributed for* $i = 1, 2, \ldots, K$.

*Proof.* This can be shown using the Jacobian change of variable formula and the density function formula for the Dirichlet distribution. For further details, see Devroye (1986). $\square$

Note that no current method exists that allows for exact sampling of general gamma variables. Furthermore, directly approximating a gamma sampling process would not naturally be differentiable with respect to the distribution parameter (now $\alpha_i$, instead of $\boldsymbol{\alpha}$). Instead, we leverage the inverse probability inverse transform theorem. The cumulative distribution function (cdf) of a $\Gamma(\alpha_i, 1)$ r.v. is $F_{\alpha_i}(x) = \frac{\gamma(\alpha_i, x)}{\Gamma(\alpha_i)}$, where $\gamma$ denotes the lower incomplete gamma function (Neuman, 2013). It does not have a well-defined inverse function, but approximations have been proposed (Knowles, 2015). We approximate $F_{\alpha_i}^{-1}(z) \approx (z \alpha_i \Gamma(\alpha_i))^{1/\alpha_i}$. Algorithm 2 describes the differentiable Dirichlet sampling approach.

