# OpenReview forum: "Towards Dynamic EHR Phenotyping: A Generative Clustering Model"
_ICLR.cc/2024/Conference — Submitted to ICLR 2024_

### Official Review · Reviewer_4mzC · 2023-10-28

**Soundness:** 3 good
**Presentation:** 3 good
**Contribution:** 2 fair
**Rating:** 3
**Confidence:** 4

**Summary:**

The authors present an innovative time-series clustering (phenotyping) approach designed to comprehensively capture both the "observational" and "outcome" dimensions of EHR time-series data. This proposed method incorporates the Dirichlet distribution under a Markovian assumption for cluster assignments and employs a VAE-like structure for forecasting future observations and predicting the outcome of interest, leveraging the cluster assignments and centroids. While distinct from prior works such as AC-TPC, T-Phenotype, and CAMELOT in temporal phenotyping, a comprehensive evaluation of the proposed method's enhancements and technical contributions (from both quantitative and qualitative perspectives) relative to existing approaches is not clear. The authors should provide a thorough exploration of the practical (clinical) validity of the experimental setup on why it is a valid scenario for the proposed method and not for the previous temporal phenotyping works. Moreover, the authors should sufficiently elucidate the implications of different components and design choices such as the number of clusters introduced in the model.

**Strengths:**

1.	The paper is generally well-written.
2.	The idea of incorporating both the outcome of interest and the observations into clustering is different from previous works.
3.	While utilizing the Dirichlet Process is not new in clustering (e.g., Dirichlet Process Gaussian Mixture Models), the authors effectively contextualize this well-known concept within the dynamic temporal phenotyping framework.

**Weaknesses:**

1.	The distinction of this work from AC-TPC and T-Phenotype is not clearly articulated in Section 2. These methods are all considered "dynamic temporal phenotyping" approaches, capable of incorporating any outcome of interest at each time step, implying that these method can also consider future (e.g., step-ahead) observations and/or the clinical outcome available at the end of the sequence as their “outcomes” at each time step.
2.	The impact of utilizing both future observations and clinical outcomes is not convincingly motivated nor adequately supported by the experimental results.
3.	The description of performance metrics should assist readers in comparing clustering performance across different methods. Additionally, it is unclear how similarity (distance) is measured to compute the Silhouette Index, Davies-Bouldin Index, and Variance Ratio Criterion.
4.	Experimental results on benchmarks and baselines appear incomplete. There is a notable absence of results on AC-TPC (as mentioned in Section 4.2) and T-Phenotype, both of which can incorporate either future observations or final outcomes of interest with a simple modification. Furthermore, results on the Normalized Mutual Information (NMI) are lacking. The discriminative power of the proposed method should be compared with the same network architecture without incorporating clustering, rather than relying solely on simple ML baselines, which may not be suitable for handling time-series data.
5.	The heterogeneity of time-series EHR data, especially its multiple modalities, which motivated this work, is not thoroughly explored. There is no architectural contribution to address this issue, and the modeling of future outcomes with a Gaussian distribution may not be suitable for binary/categorical observations, which are often prevalent in EHR data.

**Questions:**

1.	Regarding Weakness #2: What clusters were discovered in the experiments, and how do they differ from those in previous works? How do the discovered clusters vary when future outcomes are incorporated, and what happens if they are not?
2.	Regarding Weakness #3: What similarity (distance) metric is used to compute SIL, DBI, and VRI? Are the ground truth future observations taken into account?
3.	The discriminative power is relatively small, and it is not clearly stated how well the evaluated methods and the proposed method handle imbalanced labels. The authors should also provide performance at each class level and AUPRC, which would be more appropriate for assessing methods under imbalance.
4.	How was the number of clusters (K) determined? This is crucial for gauging the discriminative power of the discovered clusters.

---

> ### Author Response · Authors · 2023-11-22
> **Response to reviewer 4mzC**
>
> We thank the reviewer for their feedback. We answer some of the points raised:
>
> **[Distinction with AC-TPC and T-Phenotypes]**:     Apologies for the lack of clarity in clarifying how DeepGC differs from AC-TPC [1] and T-Phenotypes [2].
>
> AC-TPC, a deep learning approach for identifying phenotypes across two dimensions, achieves this through an iterative process involving cluster assignment and refinement. These subprocesses utilize an outcome prediction task, enabling weight backpropagation through the entire network. Unlike DeepGC, AC-TPC is limited in its capability to a) handle data generation, b) model both observation and outcome data, and c) was specifically designed for a binary outcome task, making it less adept at handling multiclass outcome, obtaining particularly poor performance when classes are highly imbalanced [3]. Additionally, AC-TPC's dynamic phenotyping relies on adverse event occurrences during the observation temporal period (I.e, it updates its phenotypes when new data are being observed). The dynamic modeling approach employed by DeepGC effectively captures the evolving phenotype assignments of a patient over time, a feature that AC-TPC struggles to achieve. T-Phenotype, builds upon AC-TPC with Laplace Encoders, seeks to enhance the trade-off between strong performance on the outcome and observation aspects. However, it shares largely similar limitations to those with AC-TPC.
>
> **[Usefulness of leveraging Future Data]**:  We disagree with the reviewer with respect to the usefulness of future observations and clinical outcomes. The premise of our work is that there are naturally occurring phenotypes in medical cohorts based on various aspects, such as observation and outcome. Incorporating outcomes, therefore, can be invaluable in improving subgroup identification and phenotyping. This can be seen regarding clustering performance. Except for TSKM, we outperform all other benchmarks with respect to clustering observation data, including purely unsupervised models.
>
> In the medical setting, having a generative model would serve the benefit of allowing us to generate trajectories of physiological variables that change over time to demonstrate and explain the behavior of our model prediction. For example, if an individual is predicted to have a cardiac arrest in the next two days, we would like to show the clinician the deterioration of his/her EHR information that changes over time. Having the ability to generate such trajectories would be helpful in model interpretation and explanation.
>
> We have empirically demonstrated that our model is able to utilize the generated information to improve outcome prediction (see Table 1 where DeepGC outperformed CAMELOT in almost all metrics on all datasets). Furthermore, it is very important to have the ability to provide information about potential patient outcomes as it is the ultimate function that clinicians look for in a decision-support tool. We will motivate the rational of future observation and outcome prediction in our revised manuscript.
>
> **[Clustering Metrics]**: The evaluation metrics were calculated via the scikit-learn [4] package in Python 3. To account for the temporal 3D dimensionality of the input data, we represented each unique feature-time pair of values as a unique feature when evaluating clustering performance.
>
> **[Benchmarking]**: We did not include experiments for AC-TPC due to the work in [5] which showed CAMELOT outperformed AC-TPC in a slightly different task. We will include the experiments with AC-TPC and T-Phenotype in the final edited submission. We note that the top performance obtained by these two benchmarks was lower to DeepGC on both datasets at least 5% (average AUROC), 4% (average F1), 2% (average Recall), 2% (average Precision).
>
> We also agree with the reviewer that it would be important to compare our baselines with similar models that do not leverage clustering, to ensure that its advantages are properly taken care of, and to that end we illustrate in the manuscript the performance obtained from using the VRNN model.

---

> > ### Author Response · Authors · 2023-11-22
> > **Response to Reviewer 4mzC (part II)**
> >
> > **[Similarity Distance for Metric Computation]**: The SIL, VRI, and DBI metrics are all computed in the input space with the assumption of an Euclidean distance. Furthermore, we did not include “future observations” when computing the clustering performance. Both choices were completed in order to ensure we could compare with all benchmarks in a fair manner.
> >
> > **[Imbalanced Data]**: Due to the imbalanced nature of the dataset, we were mainly focused on optimising for the average F1 score on the outcome aspect. The average F1 score is, by definition, an average over all outcomes, while also being more sensitive about the data imbalance for each outcome. We considered using AUPRC, but decided against usage of this metric for 2 reasons: a) unlike binary AUROC, its values are not between [0.5, 1], with binary AUPRC being bounded by the number of outcome examples across the whole cohort, for each outcome, and b) unlike AUROC, AUPRC does not have a natural associated probabilistic interpretation.
> >
> > **[Selection of K]**: The number of clusters was determined via Occam's Razor [6] approach (same as CAMELOT), as described in the last paragraph of Section 4. We increased the number of clusters until we no longer observed a statistically significant increase in the average F1 performance of DeepGC.
> >
> >
> > **[References]**:
> >
> > [1] - Changhee Lee and Mihaela Van Der Schaar. Temporal phenotyping using deep predictive clustering of disease progression. In Proceedings of the 37th International Conference on Machine learning, pp. 5767–5777. Journal of Machine Learning Research, 6 2020.
> >
> > [2] - Yuchao Qin, Mihaela van der Schaar, and Changhee Lee. T-phenotype: Discovering phenotypes of predictive temporal patterns in disease progression. In Francisco Ruiz, Jennifer Dy, and Jan-Willem van de Meent (eds.), Proceedings of The 26th International Conference on Artificial Intelligence and Statistics, volume 206 of Proceedings of Machine Learning Research, pp. 3466–3492. PMLR, 25–27 Apr 2023.
> >
> > [3] - Henrique Aguiar, Mauro Santos, Peter Watkinson, and Tingting Zhu. Phenotyping Clusters of Patient Trajectories suffering from Chronic Complex Disease, in Machine Learning for Health (ML4H 2020) at Conference on Neural Information Processing Systems, 2020.
> >
> > [4] - Pedregosa Fabian, Gael Varoquaux, Alexandre Gramfort, Vincent Michel, Bertrand Thirion, GriselOlivier, Mathieu Blondel, Peter Prettenhofer, Ron Weiss, Vincent Dubourg, Jake Vanderplas, Alexandre Passos, and David Cournapeau. Scikit-learn: Machine learning in python. Journal of Machine Learning Research, 12:2825–2830, 2011.
> >
> > [5] - Henrique Aguiar, Mauro Santos, Peter Watkinson, and Tingting Zhu. Learning of cluster-based feature importance for electronic health record time-series. Proceedings of the 39th International Conference on Machine Learning, PMLR 162, 2022.
> >
> > [6] - William Hamilton. Discussions on Philosophy and Literature, Education and University Reform: Chiefly from the Edinburgh Review; Cor., Vindicated, Enl., in Notes and Appendices. Harper & brothers, 1861.

---

> ### Comment · Reviewer_4mzC · 2023-11-23
> **Response to the Rebuttal**
>
> I appreciate the authors for their response. While some of the responses effectively addressed my initial comments, many of them have not yet addressed my concerns. Please find the follow-up comments and clarifications below:
>
> **[Distinction with AC-TPC and T-Phenotypes]**
> - AC-TPC can also handle not only binary but also multi-class categories or continuous outcomes. The only thing that needs to be changed is the loss function depending on the outcome type. (By the way, I agree that this model may not be suitable for high imblanceness).
> - What the authors mentioned "It updates its phenotypes when new data are being observed" implies the ability of AC-TPC to perform dynamic clustering. Moreover, given any trained temporal clustering method, one can take the input sequence, $x_{1:T}$ and make a set of subseries, $x_{1:1}, x_{1:2}, x_{1:3}, \dots, x_{1:T}$, to see how the cluster assignment can be dynamically changed over time. So, I do not believe the ability of DeepGC to dynamically perform clustering is a novel contribution over the other methods.
>
> **[Usefulness of leveraging Future Data]**
> - I agree with the authors that using both future observations and clinical outcomes can be very important. What I mentioned initially in the review is to point out that the authors failed to provide an in-depth analysis of how the identified clusters can be distinguished by utilizing that information.
> - It will be impossible to apply Euclidean distance to time-series with different number of observations
>
> **[Clustering metrics]**
> - I am not inquiring about the specific package you used for computing those metrics. To utilize these metrics (without the ground-truth cluster labels), one must establish the 'similarity' between input samples, which is not straightforward for time-series data. Using Euclidean distance for a pair of time series with different numbers of observations is not feasible. Furthermore, many recent deep learning-based temporal clustering methods, including DeepGC, avoid relying on Euclidean distance in the input space. Instead, they focus on leveraging latent representations for clustering. Given these considerations, how can we consider clustering performance based on Euclidean distance in the input space as a plausible metric, especially when Euclidean distance is known to be unsuitable for time series data?
>
> **[Using AUPRC]**
> - I disagree with the authors. AUPRC is a widely used metric even though the value stays between [0, 1]. Even though the value of AUPRC is low due to the high imbalance, one can compare the value with the AUPRC of the null distribution to highlight the gain of the discriminative power.
> - The F1-score depends on how a user sets the threshold for converting continuous predictions into binary predictions. If the model is well-calibrated, one can straightforwardly use the label frequency for imbalanced data. However, different performance orders may arise if the threshold is set individually for each method based on the validation performance. So, I believe a more general way to show the discrimination performance for imbalance data is AUPRC.

---

### Official Review · Reviewer_BDLh · 2023-10-30

**Soundness:** 3 good
**Presentation:** 2 fair
**Contribution:** 2 fair
**Rating:** 5
**Confidence:** 3

**Summary:**

Motivation: EHR data lacks labeled phenotypic information, relies on unsupervised learning, hard to validate. Multi-dimensional nature also makes things harder. Previous work is limited to a time frame / not  outcome-sensitive / not end-to-end.
The authors propose DeepGC combining VAE with dynamic clustering (Markovian Dirichlet distribution) to model physiological status over time. It also also enables generation of future observation data.
Evaluation performed on HAVEN and MIMIC demonstrates good performance over other baselines in all tasks, including clustering and outcome prediction.

**Strengths:**

- Motivation is clear, and grammar + style is quite fluent
- The work appears to be generally sound, with proofs, and performs well with respect to baselines
- A good number of baselines with respect to the task
- Datasets used to evaluate models are well-known in the field of ML for healthcare

**Weaknesses:**

- Please use bolded vectors for variables that are not scalars, e.g. \bm{x} = \{x_1,x_2,...\}
- The figure is slightly pixilated, please use svg or pdf format. Additionally I found it confusing. The green solid line has no arrow, whereas the other lines do. Furthermore, the genreation of x' is not depicted.
- Paper writing needs some edits. E.g. spaces after commas, X and $X$ both being used to refer to datasets, etc.
- Eqn 1 and the following paragraph has inconsistent $<$ and $\leq$
- Lack of discussion on interpretability, visualization of patient clusters.
- No ablations performed

**Questions:**

- Is it possible to to run results from general tabular data generators, like TabDDPM (https://arxiv.org/abs/2209.15421) or PAR Synthsizer (https://docs.sdv.dev/sdv/sequential-data/modeling/parsynthesizer)?
- Can the authors include s quantitative evaluation of the different patient phenotypes discovered by the algorithm?
- Further analysis of the generated data / more ablations would be useful to see the full scope of the work. E.g. How different is the generated data from the original? Parameter tuning of the RNN?
- Why does CAMELOT display a similar performance to DeepGC on one dataset and not the others?

---

> ### Author Response · Authors · 2023-11-22
> **Response to Reviewer BDLh**
>
> Thank you to the reviewer for their comments. We address the questions and weaknesses raised as follows:
>
> **[Typos and Editing]**: Figures will be made clearer on an eventual submission, as well as better care with boldening vector objects, and editing the document for errors in writing.
>
> **[Ablation]**: We believe that any type of ablation studies on DeepGC would simmer down to benchmarking our model against benchmarks already considered. Specifically, the benchmark VRNN and its variant VRNN-GMM, both subjects of comparison, can be seen as simplified versions of DeepGC. To elaborate further, VRNN-GMM is 2-stage process (i.e., the latent space is fixed and learnt from VRNN separately, then clustering is achieved via GMM), where ours is end-to-end joint learning with the combination of outcome modelling and phenotyping. Additionally, pure VRNN lacks the phenotyping component. Our argument posits that it is the synergistic combination of outcome prediction, phenotyping, and sequential modelling that enhances the models' ability to learn improved phenotypes, sensitive to both outcome and observation aspects.
>
> **[Generative Benchmarks]**: We thank the reviewer for the suggestion to use TabDDPM and PAR Synthesizer as comparisons to evaluate the quality of our generative component. We will use these comparisons for further analysis, however, we would like to stress that generation is not our main goal. DeepGC was designed to tackle the task of phenotyping across two different aspects. The addition of a generative component results from the fact that observations closer in time to an event were more separable with respect to outcome than earlier observations. Combined with the fact that observations are seen only earlier to a certain window of time from the outcome time, we developed a generative approach to generate forward looking data to allow phenotypes to better capture the relevant outcome particularities. Furthermore, the generative feature provides explainability to clinicians for the model decision making as well as improving the outcome prediction
>
> **[Training Details]**: Hyper-parameter tuning of all models was conducted in the same fashion. We used a cross-validation approach with 5 folds, iteratively training a model on a train and validation sets for each fold. The model with the top average performance on the validation data was selected. Consequently, we trained this model on the joint training-validation data, and evaluated on a hold-out test set that was unseen throughout the whole process. We evaluated testing performance using the same set of 10 seeds.
>
> **[CAMELOT Performance]**: It is not immediately clear why Camelot performs similarly to DeepGC on the HAVEN dataset, but not so on the MIMIC data. It could potentially be due to the ratio between different outcome prevalences in the data. As an example, the ratio of the number of patients with a given outcome between the most frequent and second most frequent outcomes is around 12% on HAVEN, but closer to 21% on MIMIC. Consequently, it could be the case that DeepGC has more modelling capabilities than CAMELOT, and henceforth is better equipped to navigate this variability in outcome distributions.

---

### Official Review · Reviewer_rHc6 · 2023-10-31

**Soundness:** 3 good
**Presentation:** 3 good
**Contribution:** 2 fair
**Rating:** 3
**Confidence:** 5

**Summary:**

The paper introduces a generative model designed for patient subgrouping, namely DeepGC. This proposed method is rooted in variational inference to identify phenotypes within temporal clinical data. DeepGC accomplishes this by modeling the joint distribution among input, output, and cluster probabilities derived from observational data, leveraging this joint distribution to generate future data used for making clinical predictions. Experimental results based on two real clinical datasets (HAVEN and MIMIC-IV) demonstrate that DeepGC enhances patient subgrouping capabilities when compared to baseline models.

**Strengths:**

This paper targets EHR-based clinical risk prediction which is an important research area in machine learning for health.

**Weaknesses:**

- The paper's technical novelty appears somewhat limited. In particular, it fails to adequately address the unique challenges presented by modeling EHR data compared to other sequential data. It also does not sufficiently explain the novelty in terms of model design compared to existing sequential generative models. It's important to note that in the machine learning context, several VAE-based architectures have been proposed to handle various types of sequential data, such as images, audio, and videos [1, 2].

- The model's design lacks clear explanations or motivation. For instance, it is unclear why VAE was chosen over other generative models for modeling the EHR data distribution. Furthermore, the rationale behind modeling the cluster probability distribution using a Dirichlet distribution remains unexplained.

- Connected to the previous concerns, it's uncertain whether the improvement in prediction performance stems from modeling patient subgrouping or is merely a consequence of the neural network architecture's high capacity. In other words, it is unclear whether patient subgrouping is essential, and the paper does not explore the consequences of removing patient subgrouping from DeepGC. To clarify, why did the authors not use a VAE architecture to model the temporal data (i.e., $P(X,Y)$) directly, instead of modeling $P(X,Y,\pi)$?

- The concerns raised are somewhat substantiated by the results presented in Table 1. The model with strong patient subgrouping performance (TSKM) does not perform well on downstream prediction tasks. Additionally, DeepGC's performance falls within one standard deviation of the second-best baseline models. Consequently, the paper would benefit from statistical testing to establish the significance of the proposed method.

- Several important technical details are omitted, such as the evaluation of patient subgrouping in an unsupervised setting. Furthermore, the paper does not explain how clustering metrics like SIL, DBI, and VRI were calculated. Data statistics and processing steps are also missing, including information about which clinical features were employed and how missing values were handled.

- The paper lacks the provision of code and supplementary documentation, which would enhance clarity and reproducibility.

References:

[1] S3VAE: Self-Supervised Sequential VAE
for Representation Disentanglement and Data Generation. ICLR 2020

[2] Disentangled Sequential Autoencoder. ICML 2018.

**Questions:**

Questions in the above section.

---

> ### Author Response · Authors · 2023-11-22
> **Response to Reviewer rHc6**
>
> Thank you for your contribution and for the questions raised. We have addressed the comments below.
>
> **[Technical Novelty]**: Addressing the technical novelty and challenges related to EHR data, we do not claim to have solved EHR modelling, but we disagree with the reviewer regarding the usefulness of DeepGC given the results obtained. DeepGC obtains close to top performance across all evaluation metrics (evaluated at both the observation and outcome aspects) and all datasets. We note that both models reference by the reviewer are extremely similar in methodology to VRNN [1] (in fact, one can argue that they are a particular type of VRNN-type network), but, more fundamentally, they lack the ability to phenotype or dynamically identify clusters (i.e., clustering changes over time). Recall our main task is that of developing methods capable of dynamically identifying relevant phenotypes/clusters across a variety of aspects (in our case a temporal, and an outcome aspect).
>
> **[Model Design]**:  The rationale for adopting a generative approach in our study is grounded in empirical observations. It became apparent through experimentation that existing phenotypes encountered challenges in learning meaningful and expressive patterns across both modeling aspects, particularly in a dynamic approach. We observed that the separability of outcome sub-cohorts is bigger at time indices closer to the actual event occurrence. Consequently, we decided to leverage a component capable of generating future observation data to reduce the conflation between modeling both aspects.
>
> In the medical setting, having a generative model would serve the benefit of allowing us to generate trajectories of physiological variables that change over time to demonstrate and explain the behavior of our model prediction. For example, if an individual is predicted to have cardiac arrest in the next two days, we would like to show the clinician the deterioration of their EHR information as it changes over time. Having the ability to generate such trajectories would be helpful in model interpretation and explanation. Furthermore, we have empirically demonstrated that our model is able to utilise the generated information to improve outcome prediction (see Table 1 where DeepGC outperformed CAMELOT across both tasks on almost all metrics)
>
> Crucially, our modeling assumptions hinge on the utilization of the Dirichlet distribution, playing a pivotal role in our model's performance. To achieve dynamically changing phenotype assignments, accurately modeling the temporal evolution of cluster assignment probabilities is essential. We opted for the Dirichlet distribution due to its parametrization over the space of probability vectors of dimension K, and its useful mathematical properties, such as a closed formula for the Kullback-Leibler divergence and an approximate reparameterization scheme that enables efficient backpropagation through the distribution parameters.
>
> **[Performance Improvement]**: We believe that the observed improvement in performance is not attributable to an increase in modeling capacity. Where applicable, all Deep Learning (DL) models underwent evaluation with an identical number of layers and nodes, where applicable. In fact, a closer look into our modeling assumptions shows the modeling capacity of DeepGC is slightly inferior to that of VRNN [1] due to the difference in latent variables. Notably, the benchmark VRNN, along with its variant VRNN-GMM model, can be viewed as ablated versions of DeepGC. Both approaches do not model outcome, however, they can model the multidimensional time-series data directly via latent variables. Although VRNN-GMM also implements a clustering component, this is a 2-stage process (i.e., the latent space is fixed and learned from VRNN separately, then clustering is achieved via GMM), where ours is end-to-end joint learning with the combination of outcome modeling and phenotyping that contributes significantly to the observed performance improvements.

---

> > ### Author Response · Authors · 2023-11-22
> > **Response to Reviewer rHc6 (part II)**
> >
> > **[Comparison with Benchmarks]**:  Recall that the main goal of our paper is to develop a model capable of identifying phenotypes across a variety of distinct aspects. The comparative performance of TSKM against DeepGC across both the outcome prediction aspect and observation aspect (clustering input time-series) highlights the difficulty in modeling the 2 aspects at the same time. For example, TSKM is the only model that outperforms DeepGC over the observation aspect, and yet it performs so much more poorly on the outcome prediction aspect. The cluster evaluation metrics such as Silhouette Score, Davies-Bouldin Index and Variance Ratio Criterion are naturally biased to favor TSKM as the metrics prefer convex clusters and the direct clustering on the input observation space.
> >
> > DeepGC outperforms all other benchmarks on the observation aspect in a statistically significant fashion using a Friedman’s hypothesis test. Furthermore, in the critical evaluation metric F1, which considers the severe outcome imbalance in the data, DeepGC outperforms all benchmarks on the MIMIC dataset, while showing top performance on the HAVEN data (albeit not statistically significant improvement over CAMELOT).
> >
> > **[Clustering Metrics]**: The evaluation metrics were calculated via the scikit-learn [2] package in Python 3. To account for the temporal 3D dimensionality of the input data, we represented each unique feature-time pair of values as a unique feature when evaluating clustering performance.
> >
> > Data handling followed previous work in [3], as was mentioned in the last paragraph of Section 4.1 “Both datasets underwent processing based on clinical knowledge and empirical validation as in Aguiar et al. (2022)”, and we will clarify in a revised version we refer to pre-processing steps, feature selection and dealing with missing values. Further descriptions of the data will be added to the appendix.
> >
> > **[Code and other documentation]**:  All code and supplementary material will be shared after revision. This is to ensure that we have the finalised the code for all analyses after revision of the manuscript.
> >
> > **[References]**
> >
> > [1] - Junyoung Chung, Kyle Kastner, Laurent Dinh, Kratarth Goel, Aaron Courville, and Yoshua Bengio. A recurrent latent variable model for sequential data. In C. Cortes, N. Lawrence, D. Lee, M. Sugiyama, and R. Garnett (eds.), Advances in Neural Information Processing Systems. Curran Associates, Inc., 2015.
> >
> > [2] - Pedregosa Fabian, Gael Varoquaux, Alexandre Gramfort, Vincent Michel, Bertrand Thirion, GriselOlivier, Mathieu Blondel, Peter Prettenhofer, Ron Weiss, Vincent Dubourg, Jake Vanderplas, Alexandre Passos, and David Cournapeau. Scikit-learn: Machine learning in python. Journal of Machine Learning Research, 12:2825–2830, 2011.
> >
> > [3] - Henrique Aguiar, Mauro Santos, Peter Watkinson, and Tingting Zhu. Learning of cluster-based feature importance for electronic health record time-series. Proceedings of the 39th International Conference on Machine Learning, PMLR 162, 2022.

---

### Official Review · Reviewer_1hAY · 2023-11-04

**Soundness:** 3 good
**Presentation:** 3 good
**Contribution:** 3 good
**Rating:** 5
**Confidence:** 3

**Summary:**

The authors present a deep generative model to model longitudinal EHR data as well as to identify clusters. Their model is implemented utilizing RNNs to model the latent state and variational inference is used to learn model parameters. The authors evaluate the utility of the approach on two separate datasets, where they evaluate both clustering and outcome prediction of the model against standard benchmarks.

**Strengths:**

1. I found the paper very easy to read and follow.
2. The datasets and empirical evaluation seem reasonable albeit somewhat limited.

**Weaknesses:**

1. It's not clear to me what outcome prediction task was chosen by the authors. Also, one randomly chosen outcome prediction task seems like a somewhat incomplete evaluation.
2. I am curious why the authors chose SVM, xgboost as standard outcome prediction methods instead of say more standard time series approaches like RNN and LSTM. It would be nice to see these benchmarks.
3. For both the clustering and the outcome prediction tasks, it is not at all clear from the results that DeepGC is better or significantly better than existing models.
4. It would be nice to see some interpretation of the learned clusters in the paper to motivate why we need to cluster patients at all.

**Questions:**

See weaknesses.

---

> ### Author Response · Authors · 2023-11-22
> **Response to Reviewer 1hAY**
>
> Many thanks to the reviewer for the useful comments. We address the points raised below:
>
> **[Outcome Prediction]**: The outcome prediction task in our analysis stems directly from the clinical applications. For example, within the HAVEN ecosystem, admissions to the wards were categorized based on the occurrence of clinically significant events such as cardiac arrest (Cardiac), admission to an Intensive Care Unit (ICU), death (Death), or successful hospital discharge (Discharge). We represented outcomes in a 4-class multiclass setting, emulating clinical practice and focusing on key clinical variables of interest to practitioners. Notably, our models excluded data within 48 hours of an event/discharge to prevent learning immediate deteriorations and aimed to provide clinicians with insights into potential future events.
>
> Similarly, for the MIMIC dataset, we defined outcomes based on patient location within 48 hours post-discharge from the Emergency Department (E.D.): death (Death), successful discharge (Discharge), ICU escalation (ICU), or whether the patient remained within a ward under observation (Ward). This approach, mirroring the 4-class representation above, captures clinically relevant measures of patient severity. Our rationale for these outcome definitions is grounded in their alignment with clinical practice, ensuring the applicability and robustness of our models in real-world settings.
>
> **[Benchmarking]**: We benchmarked Support Vector Machines (SVMs), XGBoost as standard outcome prediction models due to their relevance and usage in healthcare, as well as relatively easy training and hyper-parameter training procedure. Given we are proposing a Deep Learning (DL) methodology, it is unclear what would be the best LSTM or RNN-type architecture to validate our models against. Not only this, but previous approaches have been validated against types of LSTM/RNN baselines. However, we will include this in the final manuscript for completeness.
>
> We selected Support Vector Machines (SVMs) and XGBoost as standard outcome prediction models due to their established relevance and widespread usage in healthcare. Furthermore, these models are also straightforward to train and tune, making them suitable benchmarks for our objective. Regarding comparisons with other Deep Learning (DL) models, we note this is a difficult problem as there are a huge number of architectures to select and there are yet no clearly defined benchmarks in our clinical setting. Nevertheless, previous studies have employed LSTM/RNN baselines, we acknowledge the importance of including such benchmarks for a more comprehensive evaluation and will incorporate this comparison in the final manuscript. We note, however, that we have already included some DL methods as part of our benchmarks, e.g. CAMELOT, and VRNN.
>
> **[Performance]**: We believe our quantitative evaluation of DeepGC provides evidence for its advantage over other benchmarks. Focusing on the observation aspect, assessed through clustering metrics, our model outperforms all benchmarks on the MIMIC dataset (14.2 VRI increase, 0.12 DBI decrease) and on HAVEN (0.08 SIL increase, 22.7 VRI increase, 0.44 DBI decrease), except for TSKM. To address the notable performance of TSKM within our evaluation pipeline, as mentioned in Section 5, we present a rationale: a) clustering evaluation metrics favour convex clusters, and b) our evaluation metrics operate on the input data, whereas deep learning methods cluster within a latent space. Notably, TSKM outperforms all other state-of-the-art (SOTA) methods, as well, not just our proposed model.
>
> Moreover, in terms of the outcome aspect, we surpass all benchmarks on HAVEN (0.02 average AUROC increase, 0.01 average F1 increase) and on MIMIC (0.02 average AUROC increase, 0.03 average F1 increase). Our model's performance is comparable to top benchmarks across other metrics, except for the average Precision score on MIMIC data. Importantly, we demonstrate an ability to identify relevant clusters, a capability TSKM lacks, showcasing our model's effectiveness beyond previously proposed state-of-the-art models.

---

### Meta-Review · Area_Chair_F5Q7 · 2023-12-06

**Metareview:**

This paper introduces DeepGC, a novel deep learning model, which effectively uncovers dynamic clinical phenotypes in temporal Electronic Health Record data by leveraging patient trajectories and outcomes. Utilizing a generative approach with Markovian Dirichlet distribution and VAEs, DeepGC provides insights into evolving patient health statuses, outperforming benchmarks in phenotype identification and outcome sensitivity. This paper has studied a very important question in the medical context. However, the reviewers have major concerns about the technical novelty of this work and lack of technical details. I hope the reviewers' comments will help the authors to further improve the paper.

**Justification For Why Not Higher Score:**

The reviewers were not excited enough about this work. They have various concerns as described earlier.

**Justification For Why Not Lower Score:**

N/A

---

### Decision · Program_Chairs · 2024-01-16

Reject